# Observation of anti-parity-time-symmetry, phase transitions and exceptional points in an optical fibre

Arik Bergman [1,2✉], Robert Duggan[2,3], Kavita Sharma[1], Moshe Tur[4], Avi Zadok [1✉] & Andrea Alù [2,3,5✉]

The exotic physics emerging in non-Hermitian systems with balanced distributions of gain and loss has recently drawn a great deal of attention. These systems exhibit phase transitions and exceptional point singularities in their spectra, at which eigen-values and eigen-modes coalesce and the overall dimensionality is reduced. So far, these principles have been implemented at the expense of precise fabrication and tuning requirements, involving tailored nano-structured devices with controlled optical gain and loss. In this work, anti-parity-time symmetric phase transitions and exceptional point singularities are demonstrated in a single strand of single-mode telecommunication fibre, using a setup consisting of off-the-shelf components. Two propagating signals are amplified and coupled through stimulated Brillouin scattering, enabling exquisite control over the interaction-governing non-Hermitian para- meters. Singular response to small-scale variations and topological features arising around the exceptional point are experimentally demonstrated with large precision, enabling robustly enhanced response to changes in Brillouin frequency shift.

[1] Faculty of Engineering and Institute for Nano-Technology and Advanced Materials, Bar-Ilan University, Ramat-Gan 5290002, Israel. [2] Photonics Initiative, Advanced Science Research Center, City University of New York, New York, NY 10031, USA. [3] Department of Electrical and Computer Engineering, University of Texas at Austin, Austin, TX 78712, USA. [4] School of Electrical Engineering, Tel-Aviv University, Tel-Aviv 6997801, Israel. [5] Physics Program, Graduate Center, City University of New York, New York, NY 10026, USA. ✉email: Bergman.Arik@gmail.com; Avinoam.Zadok@biu.ac.il; aalu@gc.cuny.edu

Two-level system models are widely studied in several branches of physics, as they can capture a wide range of phenomena in nature. In most scenarios of interest, their dynamics is well characterized by a $2 \times 2$ Hermitian Hamiltonian, which entails real eigenvalues and a unitary evolution. Non-conserving systems, with loss and/or gain, are instead described by non-Hermitian Hamiltonians, which generally correspond to complex eigenvalues. Real eigenvalues can arise in such systems if they obey parity-time (PT) symmetry[1], i.e., if loss and gain are suitably balanced in space. As the non-Hermiticity parameter varies, PT-symmetric systems experience a phase transition, after which their eigenvalues become complex. This broken phase arises at the exceptional point (EP) of the system, at which the two eigenvalues and eigenmodes coalesce into one[2]. The eigenvalues around the EP are extremely sensitive to perturbations in the system parameters, hence EP physics has been raising great interest in recent years[3], both from the fundamental research standpoint and in the context of various signal processing and sensing applications[4], with demonstrations in a number of physical platforms to date[5–15]. For the most part, optics-based realizations have made use of integrated nanophotonic devices and nanostructures[16–22], typically relying on coupled microresonators with careful control over resonance frequencies, gain and loss balance, and coupling strength. Integrated micro- and nanodevices are compact and provide large design freedom, but at the same time fabrication process variations make the EP singularity difficult to reach, and the post-fabrication tuning of these systems is inherently limited. Moreover, nanofabrication of speciality devices restricts the compatibility and integration of non-Hermitian systems within standard platforms.

In this work, we demonstrate precise control of the dynamics of a non-Hermitian two-level system around its EP based on the propagation of light in standard telecommunication single-mode fibres, creating an anti-PT-symmetric system, and taking advantage of the precise control of signal propagation enabled by Brillouin scattering processes. Our degrees of freedom are two continuous probe signals with precise frequency detuning, on the order of 1 MHz. The two tones are amplified or depleted by a pair of continuous counter-propagating pump waves through backward simulated Brillouin scattering (backward SBS)[23], providing careful control of gain and loss along the fibre. This process is most efficient when the frequency separation between pumps and probes equals the Brillouin frequency shift (BFS) of the fibre. The SBS interaction also controls the coupling between the two probe tones through Brillouin-enhanced four-wave mixing (BE-FWM), a process first demonstrated in liquid cells over 40 years ago[24], and extensively studied in optical fibres[25], nonlinear glass[26] and integrated silicon devices[27]. Proposed applications of these processes include optical phase conjugation[28], all-optical calculus[29], coherent acousto-optic memory[30], ultrahigh-resolution optical spectrometry[31] and sensing of temperature, strain and refractive index[32–34]. Here we take advantage of this interaction to demonstrate the dynamics of non-Hermitian systems around the EP.

Our optical fibre platform provides precise control over coupling strength, frequency separation between the probe tones and detuning from the BFS of the fibre, ideally suited for our demonstration of precise tuning of the system parameters around its EP. Quite remarkably, all employed components in our experiments are readily accessible in most optics labs, and no fabrication is necessary. When the frequency detuning between the two tones and pumps' powers are properly adjusted, we are able to observe a phase transition between anti-PT-symmetric and broken-symmetry regimes, passing through an EP, clearly demonstrating coalescence of the eigenvalues. Close to the EP singularity, the system shows enhanced spectral response to

small-scale variations in the BFS. These changes may be used to resolve extremely small changes in temperature and axial strain in the fibre[35,36], up to temperature changes of 0.1 degrees Kelvin or 2 ppm of axial strain in the fibre, since the BFS at telecommunication wavelengths increases by approximately 1 MHz per degree Kelvin[35] or 20 ppm of axial strain[36,37]. Our fibre platform also provides an exciting opportunity to study the topological features of non-Hermitian systems around EPs as we intentionally detune the frequency separation between the probe tones and the offset between probes and pumps from the BFS. The topology of non-Hermitian degeneracies has drawn significant attention in recent years[38], triggered by the discovery of an asymmetry in the Riemann surfaces around the EP[39,40]. Adiabatic dynamic variations of the system parameters encircling an EP lead to asymmetric energy transfer[41] and mode switching[42]. Similarly, stationary eigenmodes have also been shown to accumulate a geometric phase after encircling an EP[43]. In our system, non-trivial topology arises in the plane spanned by the offset from the BFS and the ratio between coupling and frequency detuning of the two tones. Based on these principles, we demonstrate significant linewidth squeezing and the emergence of a discontinuity in the eigenmode spectra as we cross the EP.

## Results

**Principle of operation**. The process at the basis of our findings is illustrated in Fig. 1a. A pair of comparatively strong continuous pump waves is launched into one end of a standard single-mode fibre [point (2) in the figure]. The two tones are of equal power $P_\mathrm{p}$, and they are detuned by a frequency difference $\Delta \nu \sim 1$ MHz, which can be precisely controlled. A pair of continuous optical probe tones is launched into the opposite end of the same fibre [point (1)]. The probe frequencies are separated by the same frequency difference $\Delta \nu$. The frequency of the upper (lower) pump tone is set to be higher than the one of the upper (lower) probe tone by the BFS $\nu_B$ of the fibre [see Fig. 1b; in the figure, variations around $\nu_B$, relevant for the following discussion, are marked with $\Delta \nu_B$]. This choice leads to strong SBS interactions between pumps and probes, which introduce a controlled coupling between the two probe tones. The probes' power levels are much weaker than the pumps, and the polarization states of all four waves are aligned throughout the fibre, since we consider a total length $L$ shorter than the beat length of residual linear birefringence[44] to maintain polarization alignment. We also note that compared to other Brillouin-based anti-PT-symmetric approaches, our probe tones do not need to be spaced by the BFS[22,45], affording us great flexibility and, down the line, the freedom to add even more probes.

Backward SBS interactions between pumps and probes generate longitudinal acoustic waves within the fibre at frequencies $\nu_B$, $\nu_B - \Delta \nu$ and $\nu_B + \Delta \nu$ (see Supplementary Information). The detuning $\Delta \nu$ is much smaller than the SBS linewidth $\Gamma_B/2\pi \sim 30$ MHz, hence all three acoustic wave components are significant. The generated acoustic waves mediate the transfer of power from the pump to the probe tones. However, since the SBS gain is only few dB, we may safely assume that the pumps stay undepleted and their power is constant along the entire fibre.

The vector $\vec{A}(z) = [A_1(z), A_2(z)]^\mathrm{T}$ contains the complex amplitudes $A_{1,2}$ of the probe tones as a function of position $z$, and the wavenumber mismatch between the pair of probe tones is $\Delta k = 2\pi n \Delta \nu / c$, where $c$ is the speed of light in vacuum and $n$ is the effective refractive index of the optical mode. The evolution of $\vec{A}(z)$ under steady-state conditions is described by $i\partial \vec{A}/\partial z = \mathcal{H}_0 \vec{A}$, with the $2 \times 2$ anti-PT-symmetric Hamiltonian

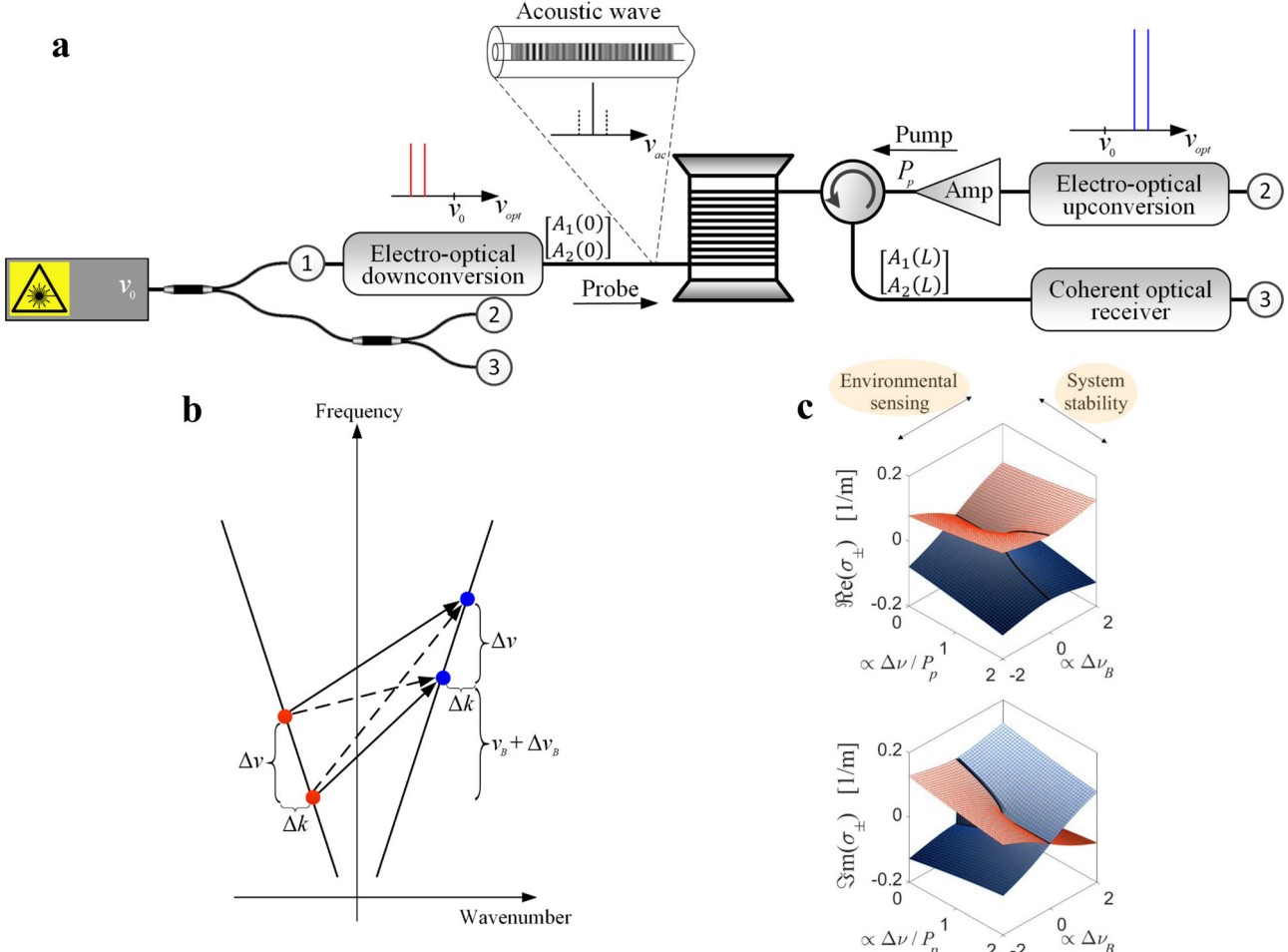

**Fig. 1 Anti-PT-symmetric Brillouin-enhanced four-wave mixing process. a** Schematic view of the fibre platform at the basis of our experiments. All optical signals are synthesized from a single, narrow-band laser diode. Electro-optic modulation is used to prepare the input probe states, as well as the SBS pumps. Coherent detection fully recovers the output probe states. **b** Phase-matching diagram of the participating signals. Both resonant and off-resonant Brillouin interactions (solid and dotted black lines) take place in the fibre. Off-resonance interactions are significant since the detuning $\Delta v$ is much smaller than the Brillouin linewidth. **c** Dependence of the probe eigenvalues $\sigma_\pm$ on the system (ratio between frequency detuning $\Delta v$ and SBS pump power $P_p$) and environmental (manifested through changes in the Brillouin resonance frequency $\Delta v_B$) conditions. The bold line refers to a system tuned to the SBS resonance ($\Delta v_B = 0$), highlighting the transition from anti-PT-symmetric to a broken-symmetry regime, through the EP (see also experimental results in Fig. 2). Measurements of $\sigma_\pm$ as a function of $\Delta v_B$ in the vicinity of the EP are shown in Fig. 3 below, and the potential use in Brillouin sensing applications is discussed further in the text.

(see Supplementary Information):

$$\mathcal{H}_0 \approx \begin{bmatrix} -\left(\Delta k + \frac{\gamma}{2}P_p\Delta\right) + i\gamma P_p & i\frac{\gamma}{2}P_p \\ i\frac{\gamma}{2}P_p & \left(\Delta k + \frac{\gamma}{2}P_p\Delta\right) + i\gamma P_p \end{bmatrix}$$

$$\equiv \begin{bmatrix} -\widetilde{\Delta k} + ig & ig/2 \\ ig/2 & \widetilde{\Delta k} + ig \end{bmatrix}. \tag{1}$$

Here, $\gamma$ is the SBS gain coefficient in units of $\mathrm{W}^{-1} \times \mathrm{m}^{-1}$ and $\Delta \equiv 2(2\pi \cdot \Delta v)/\Gamma_B$ is the normalized detuning between probe tones. The Hamiltonian includes SBS amplification of each probe tone $ig \equiv i\gamma P_p$, the coupling between probes $ig/2 \equiv i\frac{\gamma}{2}P_p$, and the total wavenumber mismatch $\widetilde{\Delta k} \equiv (\Delta k + \frac{\gamma}{2}P_p\Delta)$, which is the sum of $\Delta k$ and the additional contribution stemming from complex-valued Brillouin gain induced by the two off-resonant acoustic waves. Equation (1) is valid under the slowly varying envelope approximation and neglects linear losses, which are much smaller than the Brillouin gain in our setup. The Hamiltonian is non-Hermitian and satisfies anti-PT

symmetry[9,10,46,47], since its diagonal and off-diagonal terms obey the relations $\Im m(h_{11}) = \Im m(h_{22})$ and $h_{21} = -h_{12}^*$, respectively. The imaginary coupling is due to the joint amplification or depletion of the probes through the acoustic modes, depending on their relative phases. The eigenvalues $\sigma_\pm$ are the roots of the characteristic polynomial $\det(\mathcal{H}_0 - \sigma I) = 0$, where $I$ is the identity matrix:

$$\sigma_\pm = ig \pm \sqrt{\widetilde{\Delta k}^2 - (g/2)^2} \tag{2}$$

An EP singularity is reached when the wavenumber mismatch equals the coupling strength: $\widetilde{\Delta k} = g/2$. In our system, this requirement is met when pump power and frequency detuning are balanced:

$$\Delta v_{EP} = \gamma P_p \Big/ \left[4\pi\left(\frac{n}{c} + \gamma P_p\frac{1}{\Gamma_B}\right)\right]. \tag{3}$$

In Fig. 1c, we show the real and imaginary parts of the probe eigenvalues $\sigma_\pm$ as system parameters (ratio between frequency detuning $\Delta v$ and SBS pump power $P_p$) and environmental

conditions (manifested through changes in the Brillouin resonance frequency $\Delta\nu_B$) are varied, highlighting interesting topological features arising around the EP. In particular, we observe two interweaving Riemann surfaces centred around the EP and, when the system is tuned to the SBS resonance ($\Delta\nu_B = 0$, bold line in the figure), the eigenvalues traverse along the cuts across the Riemann sheets (black curves) as the ratio $\Delta\nu/P_p$ is varied. Adiabatically changing these parameters provides exotic topological features, which we explore and discuss in the following sections.

The BE-FWM process generally generates additional spectral sidebands of the input probe tones at frequency intervals of $\Delta\nu$[48,49], which grow as cascaded SBS processes accumulate[48,49]. Hence, a complete description of the probe propagation would require considering additional tones in (1), which makes the system higher-dimensional. However, for our system parameters and the considered fibre length, the key predictions of our simplified 2D model hold very well, as seen by our experimental demonstration of EP singularity and phase transition presented in the next section.

**Experimental demonstration of EP singularity and symmetry breaking.** Figure 2 presents the real and imaginary parts of the two eigenvalues $\sigma_\pm$ as a function of $\Delta\nu$, with $P_p = 27$ dBm, comparing their dispersion predicted by Eq. (2) (solid lines) with the retrieved data from measurements (dots). The measurement setup and protocols are described in detail in Methods, and the mean value $\bar{\sigma} = (\sigma_+ + \sigma_-)/2$ has been subtracted for convenience. For large values of detuning $\Delta\nu > \Delta\nu_{EP}$, the two eigenvalues have the same imaginary part, whereas their real parts differ. As the detuning is reduced, they coalesce at an EP singularity at $\Delta\nu_{EP} = 1.145$ MHz. Below the EP, the two eigenvalues acquire different imaginary parts, manifesting a phase transition and broken symmetry. The measured splitting of the eigenvalues agrees very well with our analytical predictions, with discrepancies in the order of $0.001$ m$^{-1}$ that may be attributed to various issues: residual imbalance between power levels of the two pump tones, the onset of cascaded SBS processes, small-scale polarization variations affecting the Brillouin gain and coupling strength, and temperature instabilities modifying the BFS. The eigenvalues exhibit drastically enhanced sensitivity to variations in $\Delta\nu$ in the EP proximity: the difference between imaginary (real)

parts of $\sigma_\pm$ scales with $\sqrt{|\Delta\nu - \Delta\nu_{EP}|}$ when $\Delta\nu$ approaches $\Delta\nu_{EP}$ from below (above), in agreement with the predictions in other photonic platforms. Further away from the EP, the splitting of eigenvalues approaches the linear dependence $|\Delta\nu - \Delta\nu_{EP}|$, as expected. The onset of the EP, the phase transition and square-root-like dispersion around the EP are all remarkably retrieved in our experiments. Compared to previous experiments in other platforms, the simplicity of our layout and the robust control enabled by SBS processes allow us to resolve the dynamics around the EP with very sharp resolution.

**Enhanced sensitivity to changes in Brillouin frequency shift.** The enhanced response to perturbations of the system parameters in the vicinity of the EP, as observed in Fig. 2, can also help detect small changes in the BFS $\nu_B$. Consider a frequency offset between the pairs of pump and probe tones that is modified from precisely $\nu_B$ to $\nu_B + \Delta\nu_B$ (see also Fig. 1b), where $\Delta\nu_B$ is on the order of a few MHz or less. When the normalized detuning is sufficiently small, $\Delta_B \equiv 2(2\pi \cdot \Delta\nu_B)/\Gamma_B \ll 1$, the Hamiltonian of the modified system can be approximated as (see Supplementary Information):

$$\mathcal{H} \approx \mathcal{H}_0 + \frac{g}{2}\Delta_B \begin{bmatrix} 2 & 1 \\ 1 & 2 \end{bmatrix}, \qquad (4)$$

with eigenvalues:

$$\sigma_\pm \approx ig + g\Delta_B \pm \sqrt{\widetilde{\Delta k}^2 - (g/2)^2 + ig^2\Delta_B/2}. \qquad (5)$$

For small $\Delta_B \neq 0$, the EP singularity cannot be reached using pump waves of equal powers, but when the pump power $P_p$ and probe pair detuning $\Delta\nu$ satisfy the EP condition of the unperturbed system (3), the difference between the two eigenvalues is proportional to $\sqrt{\Delta_B}$:

$$\Delta\sigma(\Delta\nu_B) \equiv (\sigma_+ - \sigma_-)/2 \approx \sqrt{i/2}\,g\sqrt{\Delta_B}. \qquad (6)$$

Therefore, the system in the vicinity of its EP is extremely sensitive not only to small changes in $\Delta\nu$, but also to changes in the BFS $\Delta\nu_B$, with similar square-root dependence. The splitting also scales with the Brillouin gain coefficient, in similarity to the SBS gain in uncoupled interactions. Changes to the BFS would be easier to observe in systems with larger Brillouin gain, such as highly nonlinear waveguides[26,27,30,48]. Even greater sensitivity may be obtained with higher-order EPs, arising when considering a larger number of probe tones within the same experimental platform.

Figure 3 shows our observation of enhanced sensitivity to BFS variations in the vicinity of the EP, for a pump power $P_p = 28$ dBm. The system eigenvalues $\sigma_\pm$ were retrieved for different values of $\Delta\nu$ and of the frequency separation between pump and probe pairs, yielding a maximum sensitivity for $\nu_B = 10.7583$ GHz $\pm 50$ kHz and probe detuning $\Delta\nu_{EP} = 1.31\pm0.01$ MHz. The EP shifts up relative to the value obtained in the experiment of Fig. 2 due to the higher pump power used in the current experiment. The residual uncertainty in the estimate of $\nu_B$ corresponds to variations in temperature of $\pm0.05$ degrees Kelvin[35]. The uncontrolled laboratory temperature may well have changed over a comparable range during the data acquisition time of about 5 minutes, hence even smaller changes in BFS may be observed in a thermally stable environment. Likewise, a reduction of experimental uncertainty in $\Delta\nu_{EP}$ below 10 kHz would require stabilization of the SBS gain within 0.1%. This condition is difficultly met in the open-loop operation of our platform. The observed precision of BFS measurements shown here therefore represents a lower bound on the attainable sensitivity in better controlled environments.

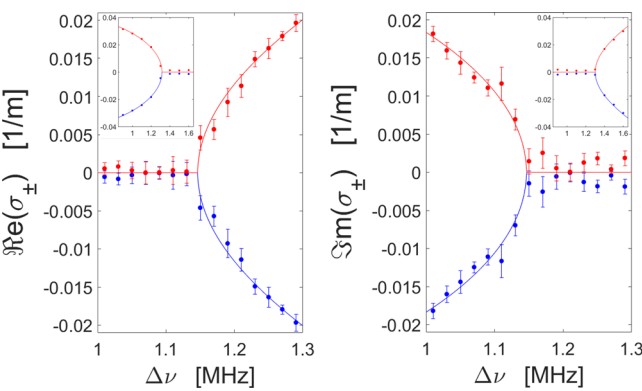

**Fig. 2 EP singularity and anti-PT-symmetry breaking.** Measured (dots) and theoretical (solid lines) $\Re$e (left) and $\Im$m (right) parts of the eigenvalues of the system. The means of the two eigenvalues have been subtracted. Each point is an average of four measurements and the error bars denote one standard deviation. The phase transition and square-root-like dispersion around the EP are evident. Insets show results over a larger, more coarse scanning range $\Delta\nu$, to highlight the transition towards linear dependence away from the EP.

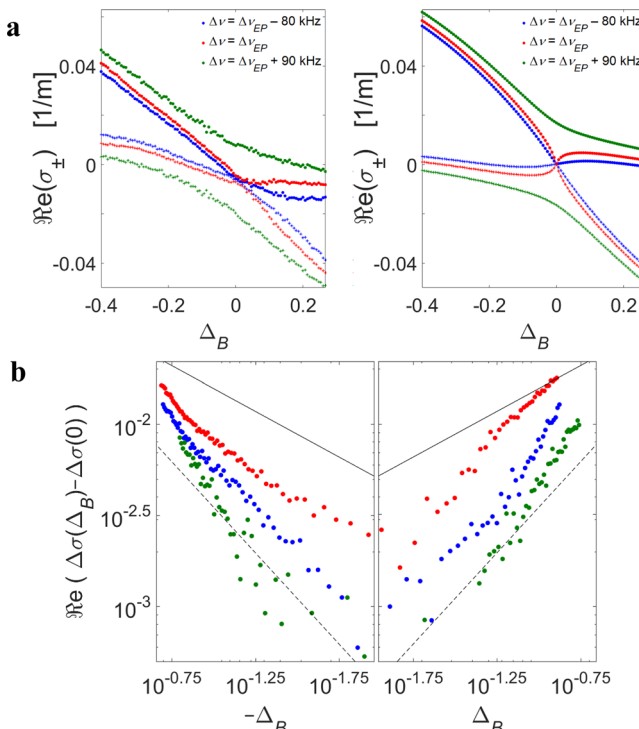

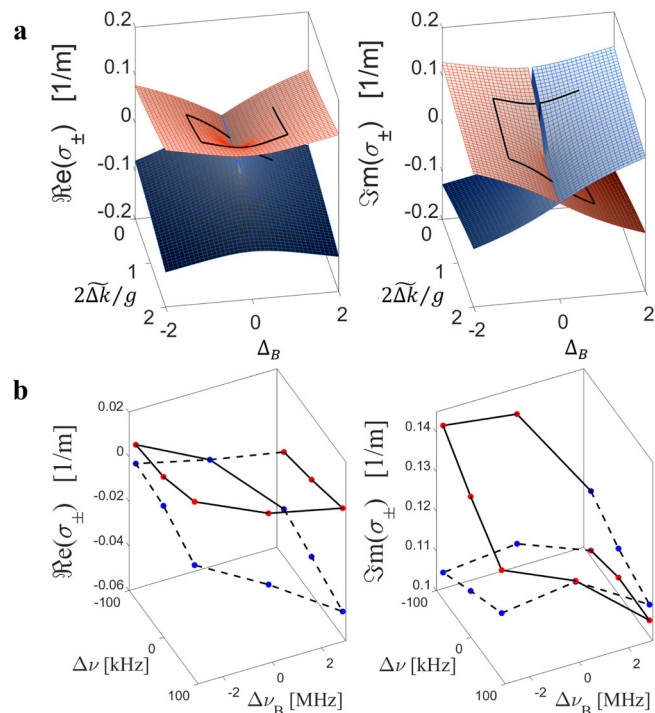

**Fig. 3 Enhanced sensitivity of the eigenvalues to deviations from the BFS at the EP.** a Measured (left) and calculated (right) $\Re$e of the eigenvalues $\sigma_\pm$ as a function of small offsets from the BFS. The frequency detuning between the two probe tones was adjusted above the EP (green, anti-PT-symmetric regime), at the EP (red), or below the EP (blue, broken-symmetry regime), see legends. The real parts of the eigenvalues repel each other at the anti-PT-symmetric regime and coalesce at the Brillouin resonance when the symmetry is broken. b Measured changes in the splitting of eigenvalues as a function of normalized detuning from the BFS: $\Re e[\Delta\sigma(\Delta_B) - \Delta\sigma(0)]$ (logarithmic scale). The colours follow the same legend as in (a). The response is the largest when $\Delta\nu$ approaches $\Delta\nu_{EP}$ (red dots). The logarithmic slope at the EP is smaller than the slopes obtained for $\Delta\nu$ below (blue dots) or above (green dots) the EP. The solid and dotted black lines mark the square root and linear dependence of the splitting on $\Delta_B/2$, respectively.

**Fig. 4 Topological encircling of EP by detuning from the BFS.** a $\Re$e (left) and $\Im$m (right) parts of the eigenvalues of the system $\sigma_+$ (red) and $\sigma_-$ (blue), as a function of the offset $\Delta_B$ from the BFS and the ratio between detuning and coupling $2\widetilde{\Delta k}/g$. The means of the two eigenvalues have been subtracted for convenience. The solid black line denotes the clockwise trajectory of $\sigma_+$, showing the swap with $\sigma_-$ after one round trip. b Traces of real and imaginary parts of the experimentally measured eigenvalues as the frequency difference between probes $\Delta\nu$ and offset of the probes from the Brillouin resonance $\Delta\nu_B$ are varied around the EP. The solid and dotted black lines denote the clockwise trajectories of $\sigma_+$ and $\sigma_-$, respectively. The eigenvalues swap after one round trip.

Figure 3a shows measured (left) and calculated (right) real parts of $\sigma_\pm$ as a function of a small offset $\Delta\nu_B$ from the estimated BFS at three different settings of probe pair detuning $\Delta\nu$. In all traces, the minimum separation $\Re e(\Delta\sigma)$ is observed near $\Delta\nu_B = 0$. In the anti-PT-symmetric regime ($\Delta\nu = \Delta\nu_{EP} + 90$ kHz, green trace), we observe an avoided crossing for the real parts of the two eigenvalues with a nonzero separation even with $\Delta\nu_B = 0$ (see also Fig. 1c). In the broken-symmetry regime ($\Delta\nu = \Delta\nu_{EP} - 80$ kHz, blue trace), and at the EP ($\Delta\nu = \Delta\nu_{EP}$, red trace), the real parts of the two eigenvalues do coalesce at $\Delta\nu_B = 0$. The slopes of the eigenvalues at the crossing point are higher when the system is at the EP, highlighting the enhanced sensitivity to $\Delta\nu_B$. This property is further illustrated in Fig. 3b, which shows on a logarithmic scale the effect of $\Delta\nu_B$ on the splitting $\Re e[\Delta\sigma(\Delta\nu_B) - \Delta\sigma(0)]$. The experimental uncertainty in the eigenvalues splitting is in the order of 0.0015 m$^{-1}$, hence smaller differences can be disregarded. The measured slope of the trace at $\Delta\nu = \Delta\nu_{EP}$ is smaller than the slopes obtained for $\Delta\nu$ below or above the EP, as expected. Enhanced response to small $\Delta\nu_B$ in the EP proximity was also found for the imaginary parts of the eigenvalues. However, a complete quantitative description of the imaginary parts of $\sigma_\pm$ on such a fine scale requires that our model is extended to account for BE-FWM generation of additional probe sidebands, beyond the zeroth-order sidebands considered in our analysis.

**Adiabatic encircling of the EP and its topological nature.** Deviations from the BFS also allow exploring the topological features associated with the EP supported by our platform. Figure 4a shows the calculated Riemann sheets of our Hamiltonian as a function of the relative probe pair detuning $2\widetilde{\Delta k}/g$ and offset $\Delta_B$ from the BFS. The surfaces representing the imaginary parts of the two eigenvalues intersect each other in the plane $\Delta_B = 0$, whereas the real parts avoid crossing. This difference between real and imaginary parts is introduced by the offset $\Delta_B$. As seen in Eq. (4), the perturbation due to $\Delta_B$ introduces real-valued coupling to the modified system Hamiltonian $\mathcal{H}$, whereas the coupling term between the tones in $\mathcal{H}_0$ is purely imaginary. As a result, the real part of the splitting between the roots of $\det(\mathcal{H} - \sigma I) = 0$ [Eq. (5)] is an even function of $\Delta_B$, while the imaginary part is odd. In addition, the imaginary part has a discontinuity at $\{\widetilde{\Delta k} < g/2, \Delta_B = 0\}$, but the real parts are equal on this line. Hence, for the eigenvalues to continuously change across this line, they have to switch their values (red surface to blue surface). These considerations do not apply for $\{\widetilde{\Delta k} > g/2, \Delta_B = 0\}$, so for one turn around the EP in parameter space of Fig. 4a we expect one swap of the eigenvalues (see the rectangular-spring-shaped black curves in the figure).

This observation is consistent with the topological nature of EPs, which manifests itself in the swapping of eigenmodes under

an adiabatic encircling of the EP in parameter space[39–42]. This encircling can be achieved in our system by varying $\Delta\nu$ and $\Delta\nu_B$. From our experimentally recovered eigenvalues, we clearly observe this topological feature in Fig. 4b. As we track the eigenvalues around the EP, we are able to experimentally observe the swap. This observation confirms that robust topological modal transfer may be possible in our fibre platform. Note that in our setup we cannot continuously change the parameters under a single excitation, as in ref. [41,42], but we retrieve the eigenvalues as we modify the driving conditions. We therefore do not directly observe handedness-dependent energy transfer, but, as evident in Fig. 4, our system nonetheless provides all the knobs to observe such a phenomenon. For instance, by introducing variations of backward SBS pumps as a function of $z$ along the fibre, probe pulses may find different pump conditions as they propagate, enabling a direct observation of topological modal transfer.

**Narrowing of eigenmode projections near the EP**. Further post-processing of output probe states reveals an additional manifestation of degeneracies in non-Hermitian operators. Enhanced response to small-scale deviations from the BFS may be observed in the eigenmode projections around the EP. In the anti-PT-symmetric region ($\Delta\nu \gg \Delta\nu_{EP}$), the output projection is Lorentzian with linewidth $\Gamma_B$. As we approach and cross the EP, however, the spectral shape changes dramatically. Figure 5 shows measurements and calculations of the output probe vector $\vec{A}(L)$, projected onto the basis spanned by the eigenmodes of the system (see Supplementary Information). We considered three distinct realizations, with $\Delta\nu = \Delta\nu_{EP} + 90$ kHz, $\Delta\nu = \Delta\nu_{EP}$ and $\Delta\nu = \Delta\nu_{EP}$ - 80 kHz, as above. In each of them, the detuning $\Delta_B$ was scanned within the range ±0.23 ($\Delta\nu_B$ between ±3.5 MHz), and the input probe vector was scanned over $N = 18$ states of the form $\vec{A}(0) = \begin{bmatrix} 1 & \exp(j\varphi_m) \end{bmatrix}^T$, $\varphi_m = 2\pi m/N$, $m = 1\ldots N$. All input states were chosen with equal magnitudes of the two tones to reduce the detrimental effects of residual nonlinearities in the optical modulators. This choice also addresses constructive and destructive interference between the two tones. The figure shows the squared moduli of the two output probe wave components $\left|A_{1,2}(L)\right|^2$ in the eigenvector basis of $\mathcal{H}(\Delta\nu, \Delta_B)$, as a function of $\varphi_m$ and $\Delta_B$.

In the anti-PT-symmetric regime ($\Delta\nu = \Delta\nu_{EP} + 90$ kHz), the projection amplitudes are enhanced with a linewidth reduced to 4.4 MHz due to the non-orthogonality of the eigenmodes. Unlike the uncoupled case, the projections become strongly dependent on the choice of $\varphi_m$ and may vary by two orders of magnitude, based on the orientation of the output state with respect to the eigenvectors. When the system operates at the EP (with an experimental uncertainty of ±10 kHz as discussed above), the output projections exhibit even more pronounced spectral squeezing, with the enhancement of projection amplitudes narrowing to a Brillouin detuning of 400 kHz. Furthermore, the projections vary by three orders of magnitude among different input phases $\varphi_m$. Below the EP ($\Delta\nu = \Delta\nu_{EP} - 80$ kHz), asymmetry appears between the two output projections with respect to $\Delta_B$, with discontinuity at the BFS. These observations are well supported by the predictions of our model and confirm the EP singularity, providing evidence of a phase transition in the eigenmode spectra and not only in the spectra of eigenvalues. Anyhow, it should be stressed that the narrowing of eigenvector basis shown here may not necessarily lead to dramatic benefits in sensing operations, since the narrowing of the eigenvector basis makes also the measurement more sensitive to noise, partially offsetting the potential advantages highlighted in Fig. 5.

## Discussion
In this work, we have implemented a non-Hermitian platform in a standard single-mode telecommunication fibre based on the propagation of two continuous probe waves, amplified and coupled through a backward SBS process. The eigenvalues of the system depend on the frequency separation between the tones and the SBS pump power. In precisely controlling these two parameters, the system can be brought from an anti-PT-symmetric to a broken-symmetry regime, through an EP. The uncertainty in measurements of the system eigenvalues is on the order of 0.001 m$^{-1}$, ten orders of magnitude smaller than the optical wavenumber. The small uncertainty enables careful observation of the phase transition and the singularities associated with the EP, beyond the limitations of other previously considered platforms for non-Hermitian physics. In particular, our experiments convincingly demonstrate enhanced response to changes in frequency detuning near the EP, and that the response

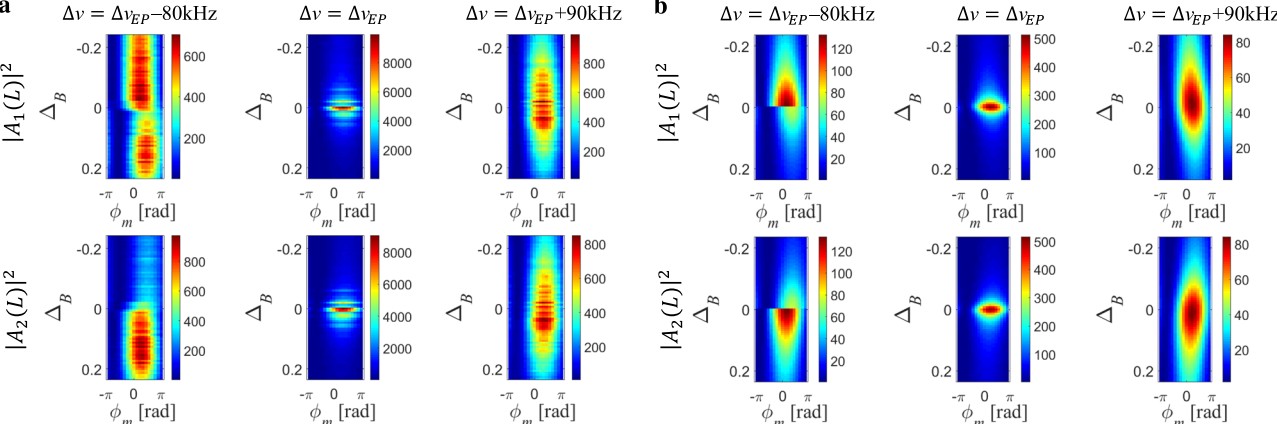

**Fig. 5 Enhanced response to small-scale variations from the BFS in eigenmode projections around the EP. a** Measured and **b** calculated squared moduli of the output probe wave projections as a function of the relative phase $\varphi_m$ between the input tones and $\Delta_B$, in the eigenmode basis of $\mathcal{H}(\Delta\nu, \Delta_B)$. The top (bottom) row presents projection on the first (second) eigenmode. The frequency detuning between the two probe tones $\Delta\nu$ was fixed above the EP (right, anti-PT-symmetric regime), at the EP (centre), or below the EP (left, broken-symmetry regime). The two output projections exhibit significant narrowing when the system approaches the EP from the anti-PT-symmetric regime. Past the EP, asymmetry appears between the two output projections with respect to $\Delta_B$, with discontinuity at the BFS corresponding to a branch-cut in the Riemann sheets.

of the eigenvalues to changes in the BFS increases near the EP as well. Our results represent the first realization of non-Hermitian physics, including phase transition and EP singularity, in a single strand of standard optical fibre. Optoelectronic systems employing fibre loops to demonstrate PT-symmetry in coupled fibre ring lasers[50], optoelectronic oscillators[51,52] and synthetic dimensions[6,53] can be found in the recent literature. However, they share many of the characteristics of PT-symmetry and EP studies in integrated devices: their degrees of freedom are optical fields circulating in resonator paths, and precise tuning of resonance frequencies is required. By contrast, the degrees of freedom in our work are co-propagating travelling waves, and no resonances are involved. Moreover, they all make use of external amplifiers to make up for coupling losses, and some use external modulators to establish coupling between the degrees of freedom. Our system, on the other hand, employs in-situ travelling-wave amplification and coupling. The accuracy, robustness and flexibility in tailoring the interaction-governing parameters are provided entirely by off-the-shelf components, and they enable the precise observation of the unusual phenomena associated with EPs and non-Hermitian physics in optics.

The two-level system model used in this work disregards the cascaded buildup of additional sidebands beyond the zeroth-order ones during propagation. While the contributions of higher-order sidebands are negligible near the input end of the fibre, they may become appreciable as SBS amplification accumulates. In our experiments, the average SBS gain of the probe tones was in the order of 5–6 dB. At this gain level, the output probe waves are expected to deviate from the predictions of our basic two-level Hamiltonian. For example, our model somewhat underestimates the experimentally observed average SBS gain. Furthermore, the additional sidebands induce differences between the imaginary parts of the eigenvalues for $\Delta\nu_B < 0$ and those observed when $\Delta\nu_B > 0$. Despite these approximations, our simplified model can fully predict all key experimental observations in our work: the transition between anti-PT-symmetric and broken-symmetry regimes through an EP and its dynamics and dispersion in parameter space. Detailed study of a four-level Hamiltonian may reveal more accurate estimations of the EP features of this fibre-optic implementation. On the other hand, a more complicated model would also make the interpretation of the results less immediate.

We have shown that the eigenmode spectra are drastically narrowed near the EP, from a linewidth of 30 MHz to just 400 kHz, with potential opportunities for Brillouin sensing applications. Measurements may be extended to a spatially distributed analysis of the BFS, using established Brillouin fibre sensing protocols[34,37,54,55]. It should be noted, however, that the enhanced response prevails only within a limited dynamic range of BFS variations, below 1 MHz. Potential sensing applications would therefore have to incorporate a closed-loop mechanism for tracking changes in the environment and in system parameters. In addition, the analysis of noise due to amplified spontaneous Brillouin scattering at the EP condition[56] leaves open questions on whether this sensing mechanism is necessarily superior to other available Brillouin sensing techniques. The sensitivity of parameter estimation near EPs has been extensively studied in recent years in numerous non-Hermitian physical platforms, exploring their fundamental and practical limitations. Some works have shown that the larger response to small-scale variations near the EP can give rise to improved sensors[57–61]. Others have however pointed out that the enhanced response may be accompanied by an elevated noise level, affecting the signal-to-noise ratio of the measurement[62–64]. In this work we have not explored this question in detail, but we envision that the large tunability of our system can support further studies of this issue.

We have also observed interesting topological features around the EP, in terms of the dependence of the eigenvalues on frequency detuning and on deviations from the BFS. The imaginary parts of the eigenvalues cross each other when the frequency difference between pumps and probes is scanned through the BFS, whereas the real parts avoid such crossing. These features may enable asymmetric energy transfer[41], mode switching[42] and accumulation of geometrical phase[43], which are now available for study over a basic fibre setup, and, in view of recent advances in Brillouin integrated photonics, can be carried over to integrated devices in order to realize configurable wavelength-selective functionalities based on broken symmetries. Overall, our experiments show how the fibre-optics platform may serve as an excellent tool to explore the exotic physics of non-Hermitian systems, PT-symmetry and EP singularities, leveraging this established platform to access and demonstrate intriguing new phenomena of great interest for basic research[65] and a broad range of applications. Nonlinear wave mixing through SBS also constitutes a suitable playground for studying the interplay between nonlinear parametric gain and PT-symmetry[66–68], and the dynamics near EPs in nonlinear non-Hermitian systems[69,70].

## Methods

**Experimental setup and procedures**. The experimental setup is illustrated in Fig. 6. All optical signals were drawn from a single narrow-band laser diode of frequency $\nu_0$ in the 1550 nm wavelength range. In the input probe path, the laser light passed through a single-sideband, suppressed-carrier electro-optic modulator (SSB-SC EOM) that was driven by the output voltage of an arbitrary waveform generator (AWG). The modulation voltage consisted of two sine waves of radio frequencies $f_{IF} \mp \Delta\nu/2$, where $f_{IF} = 5.332$ GHz is a fixed intermediate frequency. The SSB-SC EOM was adjusted to transmit the lower modulation sideband only, so that the optical frequencies of the two probe tones were $\nu_0 - f_{IF} \pm \Delta\nu/2$. The magnitudes and phases of the two input probe tones could be controlled independently by the AWG. The optical power level of each probe wave component was 3 dBm. The probe tones were launched into one end of a standard single-mode fibre under test of length $L = 10$ m.

The two pump wave components were generated in two stages. Light in the pump branch passed first through a double-sideband suppressed-carrier electro-optic modulator (DSB-SC EOM), driven by a sine-wave voltage of frequency $\Delta\nu/2$. The modulated waveform consisted of two tones of equal amplitudes and phases, at optical frequencies $\nu_0 \pm \Delta\nu/2$. The pump waves then passed through a second SSB-SC-EOM, driven by a sine-wave voltage of frequency $\nu_B - f_{IF}$. The BFS $\nu_B$ in the fibre under test was 10.762 GHz. That modulator was biased to transmit the upper modulation sideband only. The optical frequencies of the two pump waves were therefore upshifted to $\nu_0 + \nu_B - f_{IF} \pm \Delta\nu/2$. The difference between the optical frequency of the upper (lower) pump wave and that of the upper (lower) probe wave equalled the BFS $\nu_B$. The pump wave was amplified by an erbium-doped fibre amplifier and launched into the opposite end of the fibre under test through a fibre-optic circulator. The power of each pump tone was varied between 27 and 28 dBm.

Following the BE-FWM process, the output probe wave was mixed with an optical local oscillator (OLO) in a coherent photoreceiver. The OLO was drawn from the same laser diode source, and down-shifted in frequency by an offset $f_{LO} = 5.181$ GHz in another DSB-SC EOM (see Fig. 6). The polarization of the OLO was aligned for maximum interference with the output probe. The mixed signal at the coherent receiver output was detected by a balanced photo-detector of 1.6 GHz bandwidth. The coherent detection recovered the complex envelope of the output probe wave: amplitude and phase[71]. The output voltage of the receiver was sampled by a real-time digitizing oscilloscope at 2 giga-samples per second rate, and the Fourier transform of the collected trace was calculated digitally. The Fourier components at frequencies $f_{IF} - f_{LO} \mp \Delta\nu/2$ were proportional to the complex envelopes $A_{1,2}(L)$ of the two probe tones at the output of the fibre under test. All waveform generators and the sampling oscilloscope were synchronized to a common 10 MHz reference clock signal. The relative radio-frequency phases between the waveform sources and sampling oscilloscope were calibrated and adjusted.

**Data analysis protocol**. The transfer of the probe waves through the Brillouin-amplified fibre under test was characterized using the following protocol. First, the pump power $P_p$ and probe tones detuning $\Delta\nu$ were chosen and fixed. Next, the input probe vector $\vec{A}(0)$ was scanned through $N$ known states, between 8 and 18, depending on the specific experiment. The output probe vector $\vec{A}(L)$ was recorded for each input state as explained above. The $2 \times 2$ transfer matrix $\overline{\overline{M}} \equiv [\mu_{11} \mu_{12}; \mu_{21} \mu_{22}]$ between $z = 0$ and $z = L$ was estimated based on a least-squares solution to the

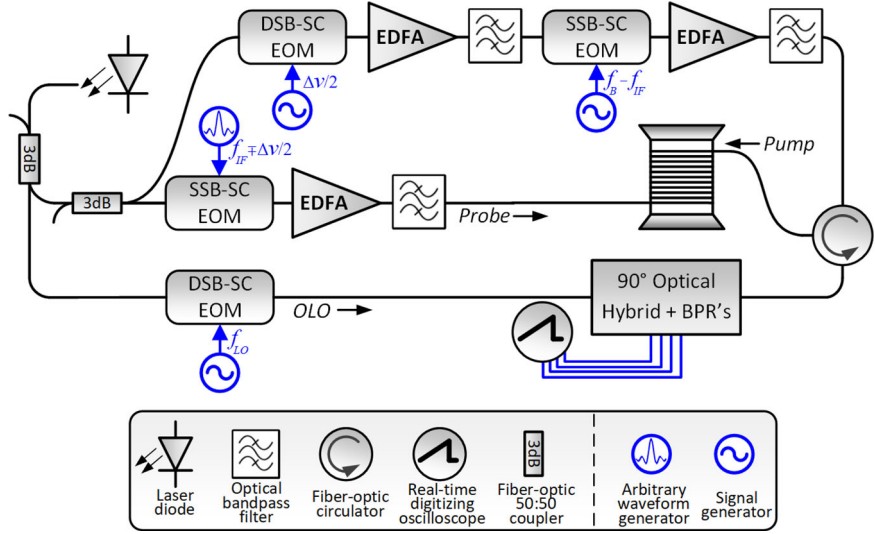

**Fig. 6 Simplified schematic illustration of the experimental setup.** S/DSB-SC EOM: single/double-sideband suppressed-carrier electro-optic modulator, EDFA: erbium-doped fibre amplifier, BPR: balanced photoreceiver, OLO: optical local oscillator.

following system of equations:

$$
\begin{bmatrix} A_{1,m}(L) \\ A_{2,m}(L) \\ \vdots \end{bmatrix} = \begin{bmatrix} A_{1,m}(0) & A_{2,m}(0) & 0 & 0 \\ 0 & 0 & A_{1,m}(0) & A_{2,m}(0) \\ & & \vdots & \end{bmatrix} \begin{bmatrix} \mu_{11} \\ \mu_{12} \\ \mu_{21} \\ \mu_{22} \end{bmatrix}, \quad m = 1 \dots N
$$

(7)

The eigenvalues and eigenvectors of the system are not observed directly on the oscilloscope. They were retrieved by first bringing the matrix $\overline{\overline{M}}$ to a diagonal form $\overline{\overline{D}}$. The two eigenvalues of $\overline{\overline{D}}$ are related to those of $\mathcal{H}$ by $\exp(-i\sigma_{\pm}L)$, assuming the fibre is uniform. The procedure was repeated for different settings of $\{\Delta\nu, \Delta\nu_B, P_p\}$.

## Data availability
The data that support the findings of this study are available from the corresponding authors upon reasonable request.

## Code availability
The codes used to process the data presented in this study are available from the corresponding authors upon reasonable request.

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

## Acknowledgements

This work was partially supported by the Office of Naval Research and the Department of Defense through a Vannevar Bush Faculty Fellowship.

## Author contributions

A.A., A.B., and R.D. conceived the idea; A.B., A.A., and A.Z. initiated the project; R.D. and A.B. developed the theory and performed the numerical analysis; A.B., A.Z., and M.T. designed the experimental setup; A.B. and K.S. constructed the experimental setup; A.B. and K.S. collected the experimental data; A.B. analysed the data; A.B., A.Z., and A.A. wrote the initial draft of the paper; all authors revised, edited and commented the paper; A.A. and A.Z. supervised and managed the project.

## Competing interests

The authors declare no competing interests.
