## [Peer Review File · Nature Communications]

Reviewers' Comments:

Reviewer #1:

Remarks to the Author:

In the paper, the authors demonstrated the implementation of anti-PT symmetry between SBS probe lights in a fiber-optic system. A detailed analytical study is presented. The response of the system near the exceptional point is studied, with high resolution by adjusting the frequency detuning between the probe lights and the SBS pumps and the power of the SBS pumps, which shows a good agreement between the theoretical and experimental results. Overall, it is a well-written paper that can be accepted for publication in Nature Communications, given that the following concerns are addressed satisfactorily.

1. With various non-Hermitian physics presented in the paper, the results seem well-predicted by the existing non-Hermitian physics, e.g., the eigenvalue bifurcation and encircling of the EP. While those phenomena are already demonstrated in various physical systems, the potential applications of such a system are only briefly discussed in the Conclusion section. The fundamental contribution of this work is rather limited. Developing a new (fiber-based) platform for APT research is useful only in a technical way. I wonder if the significance of the work can be emphasized in terms of fundamental physics or its potential in future technology.

2. The authors claim that the work is the first realization of non-Hermitian physics in optical fiber. However, a few reports about fiber lasers based on PT symmetry have been published, for example:

- a. Science 03 Jan 2020, Vol. 367, Issue 6473, pp. 59-64
- b. Light: Science & Applications Vol. 9, Article number: 169 (2020)
- c. Photonics Research Vol. 6, Issue 4, pp. A18-A22 (2018) • <https://doi.org/10.1364/PRJ.6.000A18>

3. An oscilloscope is used to sample the probe lights from the fiber link. More details may be provided to show how the real and the imaginary parts of the eigenvalue of the system are decoded by the Fourier transform of the sampled signal. The time domain signal along with its spectrum, which shows the detuning of eigenfrequency and variation of the power of the eigenfrequency, can be useful to visualize the behavior of the eigenvalue of the system as the parameters are adjusted. The signal processing details may be added in the Method Section or in the Supplementary Information.

Reviewer #2:

Remarks to the Author:

The work by Bergman et al. describes an experimental realization of exceptional points (EP) by implementing an anti-PT symmetric Hamiltonian in a standard fiber optics. In their setup, nonlinear acousto-optical interactions provide the source of non-Hermiticity. The ability to tune the system is accomplished by controlling the pump beams. By using this platform, the authors demonstrate several EP-related features including sensitivity to perturbation and Riemann sheet topology.

While several other implementations for EPs exist, this work represents an experimental milestone because: (1) It employs standard optical components; (2) The operation point can be tuned by adjusting the pump without the need to change the system itself. This is in contrast to other platforms that require daunting fabrication efforts for each operating point.

The manuscript is well-written, timely and will add an important toolbox for investigating non-Hermitian systems. I recommend it for publication after the author considers the following minor comments:

1. There are a few typos across the manuscript. For instance, in the abstract there is an extra "of"

in the sentence: "... a setup consisting of entirely of...".

2. The authors have demonstrated the square root sensitivity near an EP. Can they comment on whether this feature can be useful for sensing applications or not?

3. There are previous works that investigated the interplay between nonlinear interactions and EPs. It would be probably useful to cite some of this work: Optics letters 40, 5086-5089 (2015), Optics letters 40, 4575-4578, (2015), New Journal of Physics 18, 125006 (2016).

NCOMMS-20-30768A-Z: Authors Reply to Reviews

We would like to thank the two reviewers and the editor for their efforts and helpful suggestions. We are glad to learn that both reviewers are supportive of the publication of our work. We addressed all comments of both reviewers in the enclosed revised manuscript, as detailed below. Changes made in the manuscript are highlighted in yellow.

Response to Reviewer #1:

Comment: “In the paper, the authors demonstrated the implementation of anti-PT symmetry between SBS probe lights in a fiber-optic system. A detailed analytical study is presented. The response of the system near the exceptional point is studied, with high resolution by adjusting the frequency detuning between the probe lights and the SBS pumps and the power of the SBS pumps, which shows a good agreement between the theoretical and experimental results. Overall, it is a well written paper that can be accepted for publication in Nature Communications, given that the following concerns are addressed satisfactorily.

Reply: We thank the reviewer for his/her support of our work.

1. With various non-Hermitian physics presented in the paper, the results seem well-predicted by the existing non-Hermitian physics, e.g., the eigenvalue bifurcation and encircling of the EP. While those phenomena are already demonstrated in various physical system, the potential applications of such a system is only briefly discussed in the conclusion section. The fundamental contribution of this work is rather limited. Developing a new (fiber-based) platform for APT research is useful only in a technical way. I wonder if the significance of the work can be emphasized in terms of fundamental physics or its potential in future technology.”

Reply: Our work makes several important contributions. We have demonstrated anti-PT symmetry and EP physics principles through control over frequency degrees of freedom with in-situ parametric amplification and coupling. We were able to resolve the bifurcation of eigen-values around the EP with high accuracy. In addition, the introduction of anti-PT symmetry and EP physics to standard fiber that is accessible to a wide research community, as opposed to specialty platforms, opens exciting opportunities to explore the exotic physics of non-Hermitian systems. Lastly, owing to the recent advances in Brillouin integrated photonics, the principles we demonstrated here can be carried over to integrated devices to realize configurable wavelength-selective functionalities based on broken symmetries. We address this potential in the Discussion of the revised manuscript.

Comment: “2. The authors claim that the work is the first realization of non-Hermitian physics in optical fiber. However, a few reports about fiber lasers based on PT symmetry have been published, for example:

a. Science 03 Jan 2020, Vol. 367, Issue 6473, pp. 59-64

b. Light: Science & Applications Vol. 9, Article number: 169 (2020)

c. Photonics Research Vol. 6, Issue 4, pp. A18-A22 (2018) •<https://doi.org/10.1364/PRJ.6.000A18> [doi.org]

Reply: We thank the reviewer for addressing these important recent demonstrations of PT-symmetry in coupled fiber loops. We refer to these works in the revised manuscript. The above references share many of the characteristics of PT-symmetry and EP studies in integrated devices: the degrees of freedom are optical fields circulating in resonator paths, and precise tuning of resonance frequencies is required. By contrast, the degrees of freedom in our work are co-propagating traveling waves, and no resonances are involved, with great advantages. Our system also offers great flexibility in the tuning of the available parameters. We added a comparison, highlighting the novelty of our work, in the revised manuscript.

Comment: “3. An oscilloscope is used to sample the probe lights from the fiber link. More details may be provided to show how the real and the imaginary parts of the eigenvalue of the system are decoded by the Fourier transform of the sampled signal. The time domain signal along with its spectrum, which shows the detuning of eigenfrequency and variation of the power of the eigenfrequency, can be useful to visualize the behavior of the eigenvalue of the system as the parameters are adjusted. The signal processing details may be added in the Method Section or in the Supplementary Information.”

Reply: Our experimental setup is mapping the complex envelopes of the two probe-wave degrees of freedom to the complex magnitudes of radio-frequency components at the coherent detector output. Therefore, the magnitudes and phases of the output probe waves are identified directly. Technically, the Fourier transform of the detected signal is calculated through sampling and off-line digital signal processing. The eigenvalues and eigenvectors of the system are not directly observed on the oscilloscope. Instead, we retrieve the 2x2 transfer matrix of the system by launching known pairs of input probe fields, and measuring the corresponding outputs. The process is repeated for a large number of pairs to improve accuracy. Once the matrix is obtained its eigenvalues are calculated offline. **We added these clarifications in the revised manuscript.**

Response to Reviewer #2:

Comment: “The work by Bergman et. Al. describes an experimental realization of exceptional points (EP) by implementing an anti-PT symmetric Hamiltonian in a standard fiber optics. In their setup, nonlinear acousto-optical interactions provide the source of non-Hermiticity. The ability to tune the system is accomplished by controlling the pump beams. By using this platform, the authors demonstrate several EP-related features including sensitivity to perturbation and Riemann sheet topology.

While several other implementations for EPs exist, this work represents an experimental milestone because: (1) It employs standard optical components; (2) The operation point can be tuned by adjusting the pump without the need to change the system itself. This is in contrast to other platforms that require daunting fabrication efforts for each operating point.

The manuscript is well-written, timely and will add an important toolbox for investigating non-Hermitian systems. I recommend it for publication after the author consider the following minor comments:

Reply: We thank the reviewer for his/her appreciation of our work.

1. There are a few typos across the manuscript. For instance, in the abstract there is an extra “of” in the sentence: “... a setup consisting of entirely of...”.

Reply: **We corrected the typos in the main text.**

Comment: “2. The authors have demonstrated the square root sensitivity near an EP. Can they comment on whether this feature can be useful for sensing applications or not?”

Reply: The potential for enhanced sensitivity of measurements near the EP is still being investigated by the community. Some works suggest that the larger response to small-scale variations near the EP would give rise to improved sensors. Others claim that enhanced response is accompanied by elevated noise sensitivity, which leaves the signal-to-noise ratio unchanged. We did not carry out sensing experiments as part of this work. The large tunability of our system can support further studies of this issue. **We address this potential in the Discussion section of the revised manuscript.**

Comment: “3. There are previous works that investigated the interplay between nonlinear interactions and EPs. It would be probably useful to cite some of this work: *Optics letters* 40, 5086-5089 (2015), *Optics letters* 40, 4575-4578, (2015), *New Journal of Physics* 18, 125006 (2016).”

Reply: We thank the reviewer for bringing up these references of other non-Hermitian nonlinear parametric processes. **We address them in the revised manuscript.** We stress that there are several differences between these studies and our present work. First and foremost, the three papers are theoretical and do not report experiments. Moreover, in these papers a $\chi^{(2)}$ nonlinearity is being used, which is not available in most optical media including standard fibers. Also, losses in these works are introduced intentionally, and they are restricted to either a single frequency and/or a single spatial waveguide. These conditions mandate specialty fabrication. On the other hand, our work relies on $\chi^{(3)}$ nonlinearity over a simple standard fiber, which acts on both degrees of freedom. Anti-PT symmetry is introduced through pump waves, rather than via physical or spectral design.

REVIEWERS' COMMENTS

Reviewer #1 (Remarks to the Author):

The authors have properly addressed the Reviewer's comments and made corresponding revisions in the manuscript. The novelty and the potential impact of the work have been emphasized. A minor comment on the possible applications in sensitivity enhancement, the Riemann sheets in Fig. 4 shows that a PT-symmetry-enhanced sensitivity can occur when the signal frequency separation ($\Delta \nu$) is changed, rather than when Brillouin frequency (Δ_B) is changed. In an SBS-based sensor, the sensing information is usually encoded to Δ_B , which cannot get enhanced sensitivity. I suggest the authors clarify this point in the manuscript.

Reviewer #2 (Remarks to the Author):

The authors have addressed all the issues I raised and in doing so, improved the manuscript.

I do recommend it for publication.

December 9, 2020

NCOMMS-20-30768B: Authors Reply to Reviews

Response to Reviewer #1:

Comment: “The authors have properly addressed the Reviewer's comments and made corresponding revisions in the manuscript. The novelty and the potential impact of the work have been emphasized. A minor comment on the possible applications in sensitivity enhancement, the Riemann sheets in Fig. 4 shows that a PT-symmetry-enhanced sensitivity can occur when the signal frequency separation (δ_ν) is changed, rather than when Brillouin frequency (δ_B) is changed. In an SBS-based sensor, the sensing information is usually encoded to δ_B , which cannot get enhanced sensitivity. I suggest the authors clarify this point in the manuscript.”

Reply: We thank the reviewer for his/her suggestions. In addition to the square-root-like dispersion of δ_ν , our system also demonstrates enhanced spectral response to variations in the Brillouin frequency shift in the vicinity of the EP, although within a limited dynamic range of ~ 1 MHz. We show these results in Figure 3 in the main text.

To emphasize these characteristics, we highlighted the axis titles in Figure 1(c), and added additional clarification in the caption.

Response to Reviewer #2:

Comment: “The authors have addressed all the issues I raised and in doing so, improved the manuscript. I do recommend it for publication.”

Reply: We thank the reviewer for his/her helpful comments and support of our work.